# Reduced DAXX Expression Is Associated with Reduced CD24 Expression in Colorectal Cancer

**DOI:** 10.3390/cells8101242

**Published:** 2019-10-12

**Authors:** Ya-Chun Chen, Tsung-Hsien Lee, Shu-Ling Tzeng

**Affiliations:** 1Institute of Medicine, Chung Shan Medical University, Taichung 402, Taiwan; big383838@hotmail.com (Y.-C.C.); jackth.lee@gmail.com (T.-H.L.); 2Department of Obstetrics and Gynecology, Chung Shan Medical University Hospital, Taichung 402, Taiwan

**Keywords:** colorectal cancer, DAXX, CD24, carcinoembryonic antigen, proliferation, metastasis

## Abstract

The presence of an activating mutation of the Wnt/β-catenin signaling pathway is found in ~90% of colorectal cancer (CRC) cases. Death domain-associated protein (DAXX), a nuclear protein, interacts with β-catenin in CRC cells. We investigated DAXX expression in 106 matched sample pairs of CRC and adjacent normal tissue by Western blotting. This study evaluated DAXX expression and its clinical implications in CRC. The results revealed that DAXX expression was significantly lower in the patients with the positive serum carcinoembryonic antigen (CEA) screening results compared to the patients with negative CEA screening levels (*p* < 0.001). It has been reported that CD24 is a Wnt target in CRC cells. Here, we further revealed that DAXX expression was significantly correlated with CD24 expression (rho = 0.360, *p* < 0.001) in 106 patients. Consistent with this, in the CEA-positive subgroup, of which the carcinomas expressed DAXX at low levels, they were significantly correlated with CD24 expression (rho = 0.461, *p* < 0.005). Therefore, reduced DAXX expression is associated with reduced CD24 expression in CRC. Notably, in the Hct116 cells, DAXX knockdown using short-hairpin RNA against DAXX (shDAXX) not only caused significant cell proliferation, but also promoted metastasis. The DAXX-knockdown cells also demonstrated significantly decreased CD24 expression, however the intracellular localization of CD24 did not change. Thus, DAXX might be considered as a potential regulator of CD24 or β-catenin expression, which might be correlated with proliferative and metastatic potential of CRC.

## 1. Introduction

Colorectal cancer (CRC) is the most common malignancy in the Western world. In the United States, its incidence is increasing: in 2014, approximately 14.1 new cases were diagnosed per 100,000 individuals and over 2010–2014, the CRC mortality rate was approximately 14.8 per 100,000 individuals [1]. Genetic mutations and epigenetic alterations contribute to tumor development through oncogene activation and tumor suppressor gene inactivation. Genetic mutations including adenomatous polyposis coli (APC), TP53, KRAS, and β-catenin (CTNNB1) participate in CRC carcinogenesis [2,3]. For instance, KRAS, a protooncogene, is a downstream effector from the epidermal growth factor receptor (EGFR) family, which plays a role in mitogen-activated protein kinase and phosphoinositide 3-kinase pathway activation [4]; moreover, APC inactivation is observed in 70%–80% of sporadic colorectal adenomas and carcinoma cases, and gain-of-function β-catenin mutations have been identified in nearly 50% of the colon tumors with intact APC [5]. β-catenin is a transcriptional activator, activating the expression of many different genes, enhancing T-cell factor (TCF)/lymphoid enhancer family expression, and translocating to the nucleus to combine with DNA-binding sites. The related effect reflects in the Wnt/Wingless pathway and this effect is crucial in both embryonic development and tumorigenesis [3,6,7].

It was previously reported that death domain-associated protein (DAXX) interacts with TP53, which plays a critical role in maintaining the proliferative condition of the stem cells in intestinal crypts. Thus, DAXX may inhibit cell proliferation through multiple mechanisms [8]. Moreover, DAXX has multiple functions, such as those in transcriptional regulation, apoptosis, oncogenesis, and antiviral response [9,10,11,12,13]. DAXX, a 120-kDa protein, has been identified in several yeast interaction trap systems by using various “baits” [14]. Initially, DAXX was reported to be a protein binding to the death domain of the Fas receptor in cytosol by activating JNK [13]. Because DAXX has several functions, its underlying regulatory mechanisms in CRC carcinogenesis remain largely unknown. In a previous study, we demonstrated that DAXX suppresses the transcriptional activity of TCF4 and induces G1 arrest of CRC cells. In addition, DAXX expression was lower in human colon adenocarcinoma cells than in normal colon cells [15]. 

Carcinoembryonic antigen (CEA; CEACAM5), a 180-kDa polypeptide, is produced during fetal development and functions as a cellular adhesion factor during organ formation [16]. It is also involved in both homotypic and heterotypic interactions, which result in cellular adherence and aggregation [17,18]. CEA acts as a paracrine factor, activating human fibroblasts by signaling through both STAT3 and AKT1-mTORC1 pathways, promoting their transition to the cancer-associated fibroblast phenotype, and enhancing cell migration [19]. Clinically high serum CEA levels are correlated with worsening CRC prognoses and are associated with a high probability of metastases and a significant decrease in survival rates [20,21]. In patients with CRC, the CEA level represents an independent prognostic marker of disease progression [22]. Therefore, here, we examined the correlation of DAXX expression in cancer tissue with serum CEA levels along with clinicopathological parameters in patients with CRC.

CD24 comprises a small protein core of 27 amino acids, which is heavily glycosylated with mucin-like cell surface protein; it isolates hepatocyte progenitor cells from normal adult liver cells that are able to differentiate into hepatocytes progenitor cell populations in the normal untreated mouse liver [23]. CD24 can be used as a ligand for P-selectin (CD62P), which is expressed at higher levels on activated granulocytes and is anchored to the plasma membrane by a phosphoinositol linkage in lymphocytic and myeloid cells [24]. It is considered an adhesion molecule and a characteristic indicating cancer metastasis [25,26]. In addition, CD24 is expressed during the early stages of B-cell development, when it is highly expressed in neutrophils. CD24 is absent in normal T cells or monocytes, however it is present in the cells of cancers, including gliomas, breast cancers, hepatocellular carcinomas (HCCs), esophageal squamous cell carcinomas, and CRCs [27,28,29,30,31]. By contrast, forced CD24 expression in breast cancer cells with low or no CD24 expression (CD24^−^) reduces cell proliferation [32]. Moreover, CD24^−^/CD44^+^ stem-like cells can be generated from CD24^+^/CD44^−^ cells after oncogenic Ras pathway activation in breast cancer stem cells (BCSCs) [33]. Therefore, low CD24 expression leads to BCSC proliferation and metastasis through the Ras/Raf and Ras/PI3K pathways [34]. In addition, CD24 may be regulated through Wnt signaling and may enhance CRC cells’ colony-forming ability and promote cell motility. CD24, proposed as a β-catenin target in CRC cells, is expressed in breast cancer cells [35,36]. However, the mechanism underlying this regulation remains unclear. As a nuclear protein interacting with β-catenin and TCF transcriptional complexes, DAXX suppresses the transcriptional activity of TCF4 in CRC cells [15]. To gain insights into the genetic basis of various tumor types, DAXX plays an important role in regulating CD24-initiated survival signaling in CRC. Thus, we further demonstrated the DAXX–CD24 correlation that is related to the clinical properties of CRC. Therefore, we clarified if DAXX plays a role in CRC carcinogenesis and can provide a reference for further therapeutic targets in CRC.

## 2. Materials and Methods

### 2.1. Colorectal Tissue Preparation

Colorectal tissue samples were collected from 106 patients with CRC who underwent curative surgical resection at the tissue bank by the Scientific Ethics Committee of the College of Medicine, National Taiwan University, Taipei, Taiwan over 2005–2007 (NTUH-REC No. 200711010R). Tissue samples of patients who were pathologically diagnosed as having colon adenocarcinoma were collected, frozen immediately in liquid nitrogen, and stored at −80 °C before use. All patients provided informed consent and the study was approved by the scientific ethics committees of National Taiwan University Hospital. Tumor staging was performed according to the International Union against Cancer tumor–node–metastasis (TNM) system criteria.

### 2.2. Cancer Tissue Extraction for Western Blotting

A total of 106 patients who underwent surgical resection for CRC at the National Taiwan University Hospital between January 2005 and March 2007 were studied. We collected matched sample pairs of CRC and nontumor-surrounding tissues from the 106 patients. CRC tumors were staged in accordance with the American Joint Committee on Cancer Staging and histological analyses were performed according to World Health Organization standards by pathologists. Then, total protein was measured using the Bradford protein assay (Bio-Rad, Hercules, CA, USA) and was stored at −20 °C. Furthermore, Western blotting for these matched sample pairs of CRC and nontumor-surrounding tissues was performed using the primary specific antibodies anti-DAXX (A-12; sc-7152; Santa Cruz, CA, USA), anti-CD24 (FL-80; sc-11406; Santa Cruz), and anti-β-catenin (BD Bioscience #610153).

### 2.3. Cell Culture

The human embryonic kidney cell line 293T (no. CCL-11268) and human colon cancer cell lines including HT29 (no. HTB-38) or Hct116 (no. CCL-247) were obtained from American Type Culture Collection (Manassas, VA, USA) and maintained in McCoy 5A medium (#SH30200.01, HyClone, Waltham, MA, USA) and Dulbecco modified Eagle medium (#SH300222.0, HyClone, Logan, UT, USA), supplemented with 10% inactive fetal bovine serum (#SH30071.03, HyClone, Waltham, MA, USA) and 100 U/mL penicillin–streptomycin at 37 °C, under 95% O_2_ + 5% CO_2_. Cellular suspensions were obtained using 0.5 mL of 0.5% trypsin–EDTA (#15400-054, Gibco, Waltham, MA, USA) for 1 to 2 min.

### 2.4. Transient Transfection and PCR

For creating DAXX-knockdown Hct116 cells, approximately 3 × 10^5^ cells were seeded in 3.5 cm dishes. After 24 h of culturing, the cells were transfected with DAXX or luciferase using Lipofectamine 2000 (Invitrogen, Thermo Fisher Scientific, Waltham, MA, USA), according to the manufacturer’s instructions. After overnight incubation, the cells were transferred into a fresh incomplete medium for 20 min before transfection. A DNA–Lipofectamine 2000 mixture was prepared and then added to cells. After 4 h of transfection, the cells were transferred into fresh complete medium and incubated at 37 °C. After 48 h of incubation, the cells were detected through PCR. PCR analysis was performed for CD24 and actin (housekeeping control). RNA was extracted from the transfected cells using TRIzol (Invitrogen, MA, USA), as per the manufacturer’s protocol. RNA was then reverse transcribed to cDNA using oligo-dT 18 primers by using Superscript II reverse transcriptase (Invitrogen). Appropriate dilutions of each cDNA for subsequent PCR amplification were determined with Tag DNA polymerase and 10× Tag reaction buffer (Bio-Van). Primer sequences were designed to comply with the Primer Express Operation Guide (GenScript): 5′-CCGAGAAGCTGTGCATCTACAC-3′ and 5′-CGCCTCTGGCATTTTGGA-3′ for CD24 and 5′-GCATGGGTCAGAAGGATTCCT-3′ and 5′-ACACGCAGCTCATTGTAGAAGGT-3′ for actin. All the reactions were initially denatured at 95 °C for 3 min, followed by 35 cycles at 94 °C for 10 s and then at 56 °C for 60 s on a 2720 Thermal Cycler (Applied Biosystems, Foster, CA, USA).

### 2.5. Western Blotting for Cell Line

Polyvinylidene difluoride (PVDF) membranes (Millipore) and Whatman 3 mm paper were cut into sizes that were equal to the sodium dodecyl sulfate–polyacrylamide electrophoresis (SDS-PAGE) gels. The PVDF membranes were then immersed in methanol for 1 min, ddH_2_O for 2 min, and finally transfer buffer (25 mM Tris base, 192 mM Glycine, 15% Methanol) for 5 min. After SDS-PAGE was completed, the gels and membranes were immersed in transfer buffer for 10 min. After the transfer, the membranes were first incubated in the NET blotting solution (0.15 M NaCl, 5 mM EDTA-2 Na, 50 mM Tris, 0.25% gelatin, and 20% Tween 20) at 37 °C with gentle shaking for 30 mins to block nonspecific binding. The membranes were then incubated with primary antibodies in the blotting solution at 4 °C overnight. After the membranes were washed in 1× TBST for 5 min three times, they were incubated with HRP-conjugated secondary antibodies for 50 min at room temperature. After the membranes were washed in 1× TBST for 5 min three times again, bound antibodies were detected using an chemiluminescence system (Millipore Corporation, Billerica, MA, USA), according to the manufacturer’s instructions. The primary antibodies that were used were anti-DAXX (A-12; sc-7152; Santa Cruz), anti-β-catenin (BD Bioscience #610153), and anti-CD24 (FL-80; sc-11406; Santa Cruz), and the antibodies were imaged using a biomolecular imager (LAS-1000; GE Healthcare). 

### 2.6. Immunofluorescence Staining

The cells were placed onto collagen-coated coverslips. Next, the cells were washed two times with 1× PBS for 5 min, fixed in 2% formaldehyde and then in 4% formaldehyde in 1× PBS both for 20 min, and, finally, permeabilized with 0.5% Triton X-100 for another 30 min at 37 °C. The permeabilized cells were then blocked using the NET blotting solution. After blocking, the cells were incubated with the primary antibodies overnight at 4 °C. The primary antibodies included anti-DAXX (A-12; sc-7152; Santa Cruz) and anti-CD24 (ML5; BioLegend, 311101) antibodies. Next, the cells were washed two times with 1× PBS washing for 5 min at 37 °C and were then incubated with Alexa Flour 488-conjugated AffiniPureGoat anti-mouse immunoglobulin G (Jackson, 115-545-001) and Alexa Flour 488-conjugated AffiniPureGoat anti-rabbit immunoglobulin G (Jackson, 111-545-003) for 1 h at 37 °C. The coverslips were inverted and mounted in glycerol on slides and were then sealed with nail polish for immunofluorescent microscopy (ZEISS Axioskop, Thornwood, NY, USA).

### 2.7. Cell Migration and Invasion Assays

Cell migration assay was performed using Millicell inserts (pore size, 8 μm). First, the cells containing short-hairpin RNA against luciferase (shLuci) or DAXX (shDAXX) were seeded onto a 24-well plate at a density of 4 × 10^5^ cells per well. The plates were incubated until confluent monolayers formed. The cells were seeded with a serum-starving medium (DMEM with 0.1% FBS; BD Biosciences) in the upper chamber; the cells moved toward the serum-starving medium in the lower chamber after incubation for 48 h. Finally, the cells in the upper chamber were removed, however those attached cells that migrated or invaded into the lower chamber were fixed and stained using 0.1% crystal violet. The images of stained cells with an optical density of 595 nm were taken using Molecular Devices SpectraMax M5.

### 2.8. Statistical Analysis

Statistical analysis was performed using GraphPad Prism (version 5). The differences in the frequencies of the basic characteristics, clinical parameters, and subtypes were analyzed using the chi-square test. The associations between DAXX and the other clinicopathological features were performed using Spearman correlation analysis. All data were analyzed for significant differences using the unpaired two-tailed Student *t* test. These results are presented as the means ± standard deviations (or error bars). All experiments were performed at least in duplicate and *p* < 0.05 was considered statistically significant.

## 3. Results

### 3.1. Correlation of DAXX Expression with Clinicopathological Parameters

DAXX can considerably inhibit hypoxia-induced lung cancer cell metastasis [37]. Initially, we examined the correlation of DAXX with clinicopathological parameters in patients with CRC. We obtained matched sample pairs of CRC and nontumor-surrounding tissues from 106 patients who underwent surgical tumor resection. The characteristics of the included patients are presented in Table 1. The association of DAXX expression (median = 0.62, verified through Western blotting [WB]) in 106 patients with CRC with clinicopathological characteristics, including serum CEA screening results, are presented in Table 1. The patients were divided into high and low DAXX expression groups according to the median value. Other clinicopathological variables, including sex (*p* = 0.0700), differentiation stage (*p* = 0.1274), invasion depth (*p* = 0.5139), regional lymph node (*p* = 0.7900), distant metastasis (*p* = 0.7411), lymphatic invasion (*p* = 0.5135), and venous invasion (*p* = 0.5653), were not correlated with DAXX expression. Next, we classified the CEA levels of ≤5 and >5 ng/mL as negative and positive screening results, respectively. The serum CEA levels of 85 patients with CRC were known (n = 53 and 32 in the high and low DAXX expression groups, respectively); in the high and low DAXX expression groups, 42 (42/53 = 79.2%) and 7 (7/32 = 21.9%) patients had negative CEA screening results (*p* < 0.001, Table 1).

### 3.2. Correlation of DAXX Expression with CD24 Expression

In the 85 patients with CRC, the association between CD24 expression and CEA levels was nonsignificant (rho = 0.118, *p* = 0.1028; Figure 1A). We further evaluated the correlation between DAXX and CD24 expression in clinical cancer tissue (rho = 0.360, *p* < 0.001), indicating a significantly positive correlation between the expression of these two proteins through WB in all 106 CRC matched pairs of tumor and surrounding normal tissue (Figure 1B). In addition, the same CRC samples demonstrate significantly negative correlation between the DAXX expression and β-catenin expression (rho= −0.276, *p* < 0.005; Figure 1C). In 85 patients with CRC whose serum CEA levels were known, we further revealed a significantly positive correlation between DAXX and CD24 expression in the CEA-positive subgroup (rho = 0.461, *p* < 0.005; Figure 1E), but not in the CEA-negative subgroup (rho = 0.265, *p* = 0.0658; Figure 1D). Based on the aforementioned factors, CD24 is the target of DAXX [36], the expression of which was negatively correlated with CEA levels in patients with CRC. These data indicated that DAXX may regulate the biological mechanism in CRC cells through CD24 or the β-catenin pathway.

### 3.3. Correlation of DAXX with CRC Cell Proliferation

Our previous study indicated that DAXX suppresses TCF4 transcriptional activity and induces G1 arrest in the CRC cell line Hct116 [15]. We further determined whether DAXX is involved in regulating CRC cell proliferation and migration by knocking down endogenous DAXX using shDAXX. To elucidate the effect of DAXX knockdown using shDAXX on CRC cell growth, we cultured 293T and Hct116 cells in DMEM medium for 0–60 h. A significant increase in proliferation was observed in DAXX-knockdown CRC cells compared with the negative control (luciferase-knockdown CRC cells) at 36, 48, and 60 h. The results showed that cell proliferation increased by 1.77-fold at 60 h in DAXX-knockdown Hct116 cells compared with the negative control (luciferase-knockdown CRC cells; *p* < 0.005; Figure 2A–D). Thus, DAXX knockdown significantly increased the proliferation of Hct116 cells compared with that of 293T cells.

### 3.4. Correlation of DAXX with CRC Cell Migration

DAXX suppresses EMT and cell invasion abilities by restraining Slug activity in vitro and in vivo for lung cancer [37]. Images that were captured using an inverted microscope under 200× and 100× magnification revealed invaded cells (black) on the Matrigel surface. The results showed that the DAXX knockdown significantly increased 293T and Hct116 cell migration 1.86-fold and 3.71-fold at 48 h, respectively, as determined using a Transwell assay (both *p* < 0.05; Figure 3A–C). In addition, DAXX-knockdown levels in Hct116 cells that were stably transfected with the constitutively active DAXX was analyzed through WB. Moreover, DAXX reversed cell migration in Hct116 cells 0.58-fold compared with the DAXX-knockdown cells (*p* < 0.05; Figure 3D–F). Our finding clarified that DAXX suppressed the cell migration ability in CRC cells.

### 3.5. Regulation of CD24 Expression by DAXX in CRC Cells

Several studies have shown that CD24 expression induces cancer progression in various cancers with clinical recurrence and poor prognosis [38,39]. In addition, CD24 overexpression could indicate tumor invasiveness and a prognosis marker in gastric cancer (GC) [40]. A study demonstrated that the human colorectal adenocarcinoma cell line HT-29 and Hct116 cells can be CD24-positive and CD24-negative controls for CRC, respectively [41]. We previously analyzed gene microarray expression to identify DAXX-regulated genes and found that DAXX affects CD24 expression in Hct116 cells (unpublished data). In the present study, DAXX expression decreased 0.26-fold in DAXX-knockdown Hct116 cells (*p* < 0.005; Figure 4A), whereas CD24 expression increased 1.78-fold in HT29 cells compared with the negative control of Hct116 cells (luciferase-knockdown CRC cells; *p* < 0.005; Figure 4B). According to WB and RT-PCR results, compared with HT29 cells, CD24 protein and mRNA expression was suppressed after DAXX-knockdown in Hct116 cells, respectively (both *p* < 0.005; Figure 4B,C). Studies supporting this hypothesis show that cytoplasmic CD24 inhibits ARF binding to nucleophosmin (NPM), ultimately resulting in decreased levels of p53 in prostate cancer (PC) cells [42]. A recent study shows that nuclear CD24 is associated with metastasis and poor outcomes in bladder cancer and CRC patients [43]. Through immunofluorescence assays, we further analyzed CD24 localization in colorectal cells: nuclear CD24 expression in the nucleus was regulated by shDAXX and decreased with further DAXX knockdown (Figure 4D). The results showed that DAXX did not change CD24 localization in DAXX-knockdown Hct116 cells by immunofluorescence analysis. Only nuclear staining of CD24 (green) expression was significantly decreased in DAXX-knockdown cells (shDaxx). 

## 4. Discussion

DAXX functions as either a proapoptotic or antiapoptotic factor depending on the cell type and context [44]. Loss of DAXX expression results in extensive apoptosis and embryonic lethality. These findings are contradictory to the role of DAXX in promoting Fas-induced cell death, suggesting that DAXX is an essential gene in mouse development, with an apoptosis prevention role [45]. A recent study reported that DAXX knockdown increased cell proliferation in a rat insulinoma cell line [46]. Jiao et al. identified that mutations in MEN1 and DAXX/ATRX lengthen the survival of patients with pancreatic neuroendocrine tumors (PanNETs) [47]. Similarly, the alternative lengthening of telomeres and the loss of DAXX/ATRX expression likely play a considerable role in driving distant metastases in patients with PanNETs [48]. In a few clinical studies, DAXX or DAXX/ATRX expression loss was correlated with chromosomal instability and predicted relapse in patients with low-stage cancer (i.e., stage I–III with no distant metastasis) with DNA hypomethylation regulation [11,49]. Furthermore, DAXX expression repressed IL-6 transcription through histone deacetylase 1-mediated histone deacetylation in macrophages [50]. However, DAXX interacts with transcription regulators that regularly and directly stimulate proliferation in ovarian cancer cell line ES-2 [44]. Furthermore, it is involved in various signaling pathways, such as acute promyelocytic leukemia, complex apoptosis, chromosome segregation, transcription, tumor suppression, heat shock response, and viral infection [13,51,52,53,54,55,56]. Our data indicated that DAXX is a main repressor for controlling CRC cell proliferation and migration (Figure 2 and Figure 3). Notably, no functional role linking DAXX to disease progression has ever been established. Our clinical data indicated that DAXX expression is not correlated with clinicopathological variables, such as sex, differentiation stage, invasion depth, regional lymph node, distant metastasis, lymphatic invasion, and venous invasion (Table 1). However, DAXX expression is correlated with serum CEA screening results in Table 1. 

Clinically, serum CEA levels are mainly used as a tumor marker to monitor colorectal carcinoma treatment and identify recurrences after surgical resection [57]. CEA was first reported in malignant tumors of endodermal-derived epithelium of the gastrointestinal tract and pancreas [58]. In a clinical study, preoperative serum CEA levels were relatively high in patients with distant metastases [59]. Furthermore, CEA inhibits circulating cancer cell death and binds to heterogeneous nuclear RNA-binding protein M4 and then activates Kupffer cells to secrete various cytokines that change the microenvironments for CRC cell survival in the liver [20]. Therefore, glycosylation-modified CEA, which is highly expressed in CRC, may be critical in tumor progression [16]. Moreover, CEA that is produced by colorectal cancer cells interacts with β-catenin protein [60]. In our previous study, DAXX suppresses the transcriptional activity of TCF4 in CRC cells [15]. In clinical practice, CEA levels of ≤5 and >5 ng/mL are considered to indicate negative and positive screening results, respectively. In our 85 patients with CRC with known CEA levels, CEA screening positive results were more common in the low DAXX expression group. CEA and CD24 are adhesion proteins that can have a macroscopic impact in terms of cancer progression and metastasis [16,23]. Moreover, several studies have shown that β-catenin acts as a transcriptional activator that activates downstream CD24 expression in Wnt signaling [35,36]. The WB data did not reveal the correlation of CD24 expression with serum CEA levels in CRC (rho = 0.118, *p* = 0.1028; Figure 1A). Therefore, we investigated the relationship of DAXX and CD24 expression in surgical specimens of patients with CRC. Our data indicated that DAXX expression is positively correlated with CD24 expression (rho = 0.360, *p* < 0.001) through WB (Figure 1B). DAXX and β-catenin expression were negatively correlated (rho = −0.276, *p* < 0.005; Figure 1C). Moreover, in the CEA-positive subgroup (36 of our 85 patients with CRC), DAXX and CD24 expression were significantly positively correlated (rho = 0.461, *p* < 0.005; Figure 1E). A critical finding was that DAXX is a tumor suppressor that may regulate the biological mechanism through the CD24 or β-catenin pathway in patients with CRC.

Studies have demonstrated that higher CD24 expression is associated with shorter patient survival, making CD24 a significant prognostic marker for malignant tumors. CD24 knockdown reduced EGFR levels and cell migration velocity through RhoA activity in the GC cell line SGC-7901 [61]. In addition, CD24 was overexpressed in aggressive HCC cell lines and the tumor tissues of patients with recurrent HCC based on the expression of EGR, an early growth response protein. CD24 expression was positively correlated with malignancy in HCC [62]. In the human lung carcinoma model A549, CD24 expression increased the expression of p-STAT3 (Y705), p-FAK (Y925), and p-Src (Y416) via Src, therefore it is associated with a poor prognosis of cancer [63]. However, these reports have indicated varying or conflicting results of experiments evaluating the adverse effect of exposure to CD24 expression. Interestingly, CD24 is a dynamically regulated cell surface protein [64]. Moreover, CD24 expression is directly suppressed, which reduces cell proliferation and metastasis via the Ras/Raf and Ras/PI3K pathways in NIH/3T3 cells [34]. However, the latest study shows that the altered expression of CD24 did not significantly affect the proliferation or spontaneous apoptosis of the HCT116 and HT29 cells [65]. Currently, the mechanism underlying the effect of targeting CD24 on the growth of CRC has not been clarified. Thus, we majorly investigated DAXX as an important repressor in controlling CRC cell proliferation and migration (Figure 2 and Figure 3). Taken together, DAXX downregulation attenuates CD24 mRNA and protein expression in Hct116 cells (Figure 4B,C). Similarly, in our study, DAXX expression was positively correlated with CD24 expression in malignant patients with CRC. Taken together, these findings support DAXX upregulation as a prognostic marker for carcinoma progression.

DAXX is extensively distributed in the cytoplasm and cell membrane in patients with high-risk human papillomavirus (HPV)-positive cervical cancer. By contrast, DAXX was distributed in the nucleus of normal cervical epithelial cells [66]. Therefore, we can further assume that after tumor cell invasion, DAXX reduces intracellular CD24 expression in the nucleus. Furthermore, we performed CD24 staining to demonstrate CD24 localization in tumor cells and noted various staining patterns with respect to staining intensity. However, the role of CD24 in CRC cells remains unclear and the mechanism underlying nuclear CD24 expression is not well established. Considerable evidence indicates that cell membrane-localized CD24 immunostaining was observed in well-differentiated sites with pearl formation, whereas cytoplasm-localized CD24 immunostaining was observed at tumor invasion sites without keratinized cells [29,38,67]. These findings are consistent with the fact that the number of lymph node metastases is a main factor in the TNM staging system, and cytoplasmic CD24 localization has also been observed in the transition of colorectal epithelial cells toward a more invasive phenotype that is associated with nodal metastasis [38]. However, CD24 can demonstrate early upregulation during tumorigenesis and can express this in the nucleus [68]. In this study, we first demonstrated that CD24 expression was distributed in the nucleus in DAXX-knockdown Hct116 cells through immunofluorescence analysis (Figure 4D). The analysis did not reveal any association between CD24 expression and poor prognosis (data not shown). However, these observations indicate that DAXX nuclear expression probably results from changes in protein expression, disturbance in protein distribution, or degradation within cells. Through proliferating cell nuclear antigen staining for detecting CRC tissue proliferation, the extent of the increase in DAXX expression can be detected—a possible future clinical research direction.

## 5. Conclusions

In our 106 patients with CRC, reduced DAXX expression was correlated with reduced CD24 expression. Our data further demonstrated simultaneously lower DAXX and lower CD24 expression in the specimen of CRC, which may be involved in the underlying regulatory mechanism. In Hct116 cells, DAXX knockdown significantly increased cell proliferation and metastasis abilities. Furthermore, lower DAXX attenuates CD24 protein expression in DAXX knockdown Hct116 cells. However, no direct change of CD24 localization was observed after DAXX knock-down in Hct116 cells. Therefore, in addition to correlating with serum CEA screening results, DAXX may act as a potential repressor by controlling downstream CD24 expression in patients with CRC.

## Figures and Tables

**Figure 1 cells-08-01242-f001:**
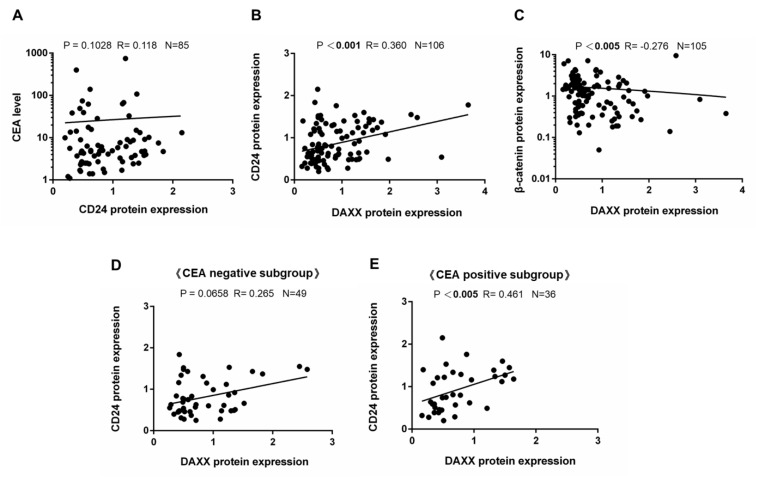
DAXX expression decreased in colorectal tumor and was correlated with CD24 expression. These protein levels were evaluated by WB in 106 matched pairs of colorectal cancer (CRC) and nontumoral- surrounding tissues. Spearman correlation analysis revealed that the correlation between (**A**) CD24 expression and CEA level was nonsignificant (rho = 0.118, *p* = 0.1028), (**B**) DAXX expression and CD24 expression was significant (rho = 0.360, *p* < 0.001), (**C**) DAXX expression and β-catenin expression was significant (rho = −0.276, *p* < 0.005), (**D**) DAXX expression and CD24 expression was significant (rho = 0.265, *p* = 0.0658) in the CEA screening-negative subgroup, and (**E**) DAXX expression and CD24 expression was significant (rho = 0.461, *p* < 0.005) when evaluated in the CEA screening-positive subgroup. β-actin was the internal control.

**Figure 2 cells-08-01242-f002:**
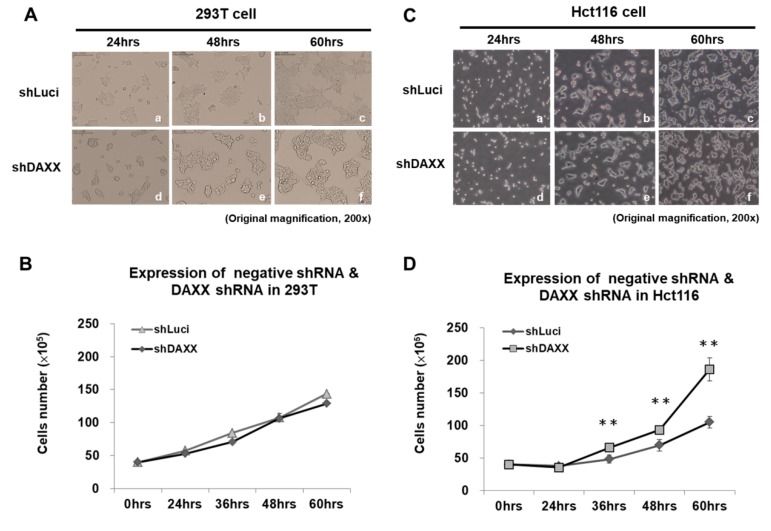
DAXX expression decreased, promoting CRC cell proliferation. We cultured (**A**) 293T (human embryonic kidney) and (**C**) Hct116 (human malignant CRC) cells for 0–60 h. Trypan blue stain assay was used to analyze proliferation rates. (**B**,**D**) DAXX knockdown promoted Hct116 cells proliferation, but not that of 293T cells. At 36, 48, and 60 h, a significant increase was noted in CRC cells proliferation after DAXX knockdown compared with luciferase-knockdown CRC cells (negative control). ** *p* < 0.005, Student *t* test; n.s. = nonsignificant.

**Figure 3 cells-08-01242-f003:**
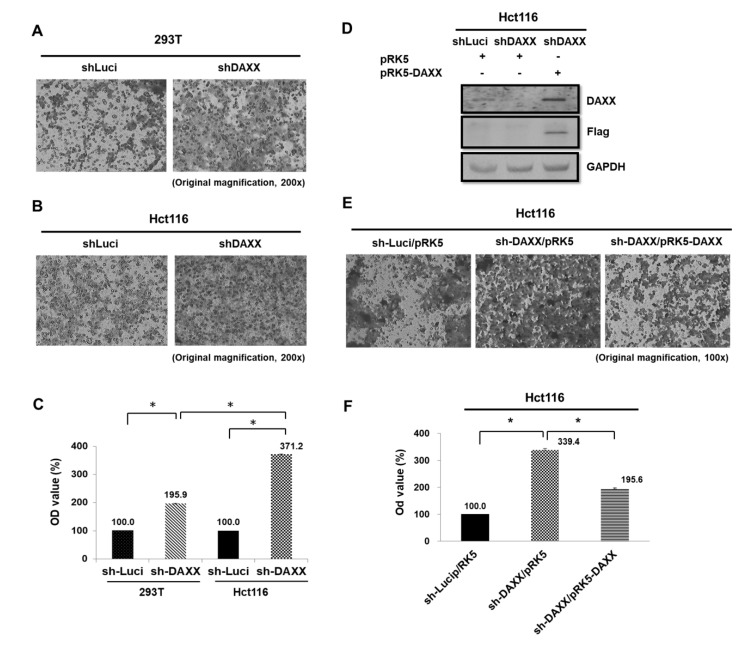
Effects of DAXX knockdown on CRC cell migration. (**A–C**) In Transwell migration assay, 4 × 10^5^ 293T cells and Hct116 cells plated onto a 24-well plate for 48 h. DAXX knockdown promoted cell motility in Hct116 cells more than in 293T cells. Luciferase-knockdown CRC cells control were evaluated as negative controls. (**D**,**E**) In addition, transfection of pRK5-DAXX into Hct116 cells of shDAXX decreased cancer cell migration so that DAXX change was reversed for cell migration in Hct116 cells. * *p* < 0.05, Student *t* test; n.s. = nonsignificant. Magnification, 200× (**A**,**B**) and 100× (**E**).

**Figure 4 cells-08-01242-f004:**
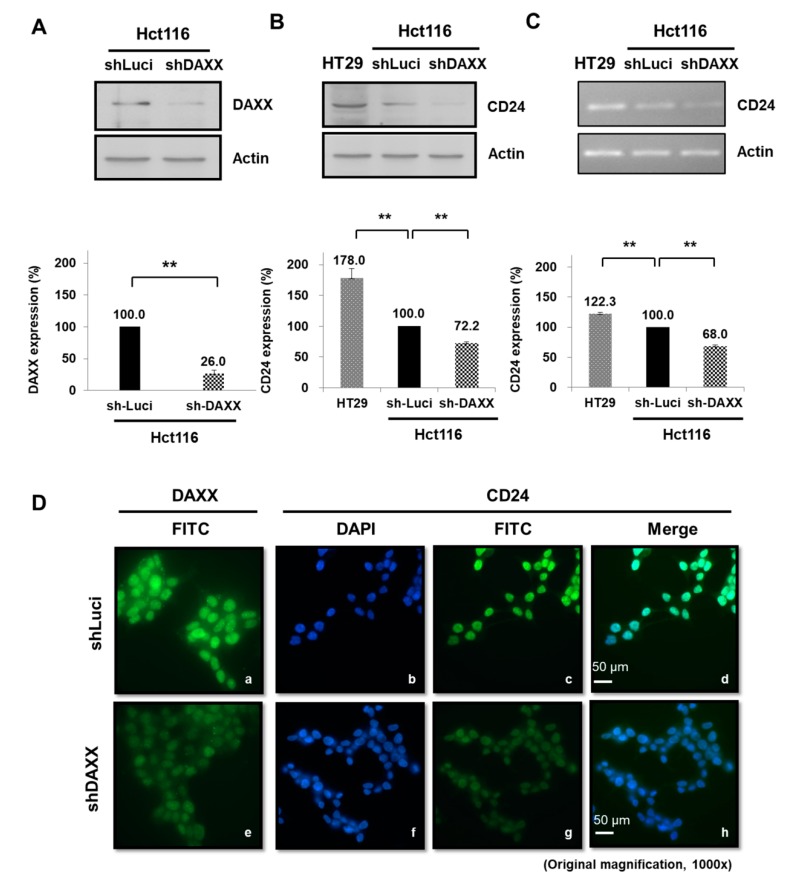
DAXX expression was low in Hct116 cells and was correlated with the downregulation of CD24 expression. (**A**) DAXX expression decreased after DAXX knockdown by using shDAXX. Luciferase-knockdown CRC cells was used as the control. CD24 (**B**) protein and (**C**) CD24 mRNA expression decreased in DAXX-knockdown cells at a level lower than luciferase-knockdown Hct116 cells (negative control) and HT29 cells (positive control) according to WB and RT-PCR results, respectively. β-actin was the internal control. ** *p* < 0.005, Student *t* test; n.s. = nonsignificant. (**D**) Immunofluorescence analysis indicated nuclear DAXX localization (green), which could be regulated using shDAXX: DAXX expression (green) decreased in DAXX-knockdown cells. CD24 localization did not change in DAXX-knockdown Hct116 cells. Only nuclear staining of CD24 expression (green) significantly decreased in DAXX-knockdown Hct116 cells. DAPI was the nuclear stain.

**Table 1 cells-08-01242-t001:** Associations between death domain-associated protein (DAXX) expression and clinicopathological characteristics of colorectal cancer patients.

				DAXX Protein Expression				
		Total		DAXX Low		DAXX High		*p* Vale
		n	%	n	%	n	%	
Parameter		106	100.0%	53	50.0%	53	50.0%	
Sex	Male	61	57.5%	35	33.0%	26	24.5%	0.0770
	Female	45	42.5%	18	17.0%	27	25.5%	
Differentiation	Well	4	3.8%	4	3.8%	0	0.0%	0.1274
	Moderate	16	15.1%	7	6.6%	9	8.5%	
	Poor	77	72.6%	40	37.7%	37	34.9%	
	Unknown	9	8.5%	2	1.9%	7	6.6%	
Depth of invasion	T1	4	3.8%	3	2.8%	1	1.0%	0.5139
	T2	1	0.9%	1	0.9%	0	0.0%	
	T3	88	83.0%	44	41.5%	44	41.5%	
	T4	3	2.9%	1	1.0%	2	1.9%	
	Unknown	10	9.4%	4	3.8%	6	5.6%	
Regional lymph node	N0	55	51.9%	29	27.4%	26	24.5%	0.7900
	N1	42	39.6%	21	19.8%	21	19.8%	
	Unknown	9	8.5%	3	2.8%	6	5.7%	
Distant metastasis	M0	54	51.0%	27	25.5%	27	25.5%	0.7411
	M1	45	42.4%	24	22.6%	21	19.8%	
	Unknown	7	6.6%	2	1.9%	5	4.7%	
Lymphatic invasion	Negative	35	33.0%	19	17.9%	16	15.1%	0.5135
	Positive	53	50.0%	25	23.6%	28	26.4%	
	Unknown	18	17.0%	9	8.5%	9	8.5%	
Venous invasion	Negative	71	67.0%	38	35.8%	33	31.2%	0.5653
	Positive	18	17.0%	11	10.4%	7	6.6%	
	Unknown	17	16.0%	4	3.8%	13	12.2%	
Serum carcinoembryonic antigen (CEA) level	Negative	49	46.2%	7	6.6%	42	39.6.%	<0.001 ^***^
	Positive	36	34.0%	25	23.6%	11	10.4%	
	Unknown	21	19.8%	21	19.8%	0	0.0%	

Correlation of DAXX expression with clinicopathologic characteristics of patients with CRC (*n* = 106) DAXX expression expressed as medians; 46.2% of the cases classified as CEA screening negative (CEA ≤5 ng/mL), 34.0% as CEA screening positive (CEA >5 ng/mL), and 19.8% as unknown. DAXX expression was significantly associated with CEA screening results (*p* < 0.001). No significant difference in other parameters. *** *p* < 0.001, chi-square test.

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
