# Peer review of "Reduced DAXX Expression Is Associated with Reduced CD24 Expression in Colorectal Cancer"

_cells, 2019, doi:10.3390/cells8101242_

Round 1
Reviewer 1 Report
The manuscript demonstrates that reduction in DAXX expression in colorectal cancer is associated with reduced CD24 expression. Daxx is a human cell death-associated protein isolated as a Tcf4-interacting protein. A reduction in Daxx protein expression has already been demonstrated in colon adenocarcinoma tissue however, its association with CD24 is the major interesting observation in the manuscript. The authors have used genetic knockdown approaches to demonstrate an association between these two proteins. Overall, the manuscript demonstrates an interesting finding using approaches which are adequate and the results support the conclusion. Some minor suggestions:
Figure 2A, the authors should provide better contrast images. Figure 3D, the loading control GAPDH is almost invisible, a better quality image is needed. Figure 4C, the PCR bands are almost invisible. A better replacement images is needed. The manuscript should be screened for typographical errors, especially in the discussion section.Author Response
Comments from Reviewer 1
The manuscript demonstrates that reduction in DAXX expression in colorectal cancer is associated with reduced CD24 expression. Daxx is a human cell death-associated protein isolated as a Tcf4-interacting protein. A reduction in Daxx protein expression has already been demonstrated in colon adenocarcinoma tissue however, its association with CD24 is the major interesting observation in the manuscript. The authors have used genetic knockdown approaches to demonstrate an association between these two proteins. Overall, the manuscript demonstrates an interesting finding using approaches which are adequate and the results support the conclusion. Some minor suggestions:
Comment 1: Figure 2A, the authors should provide better contrast images. Figure 3D, the loading control GAPDH is almost invisible, a better quality image is needed. Figure 4C, the PCR bands are almost invisible. A better replacement images is needed. The manuscript should be screened for typographical errors, especially in the discussion section.
Response 1: Thank you for your comments. We have incorporated your suggestion throughout the manuscript. But in Figure 4C, we confirm the CD24 mRNA was only specific PCR band. Moreover, the human colorectal adenocarcinoma cell line HT-29 and Hct116 cells can be CD24-positive and CD24-negative control for CRC, respectively [40]. As shown in Figure below, CD24 mRNA expressions were detected in variety human colon carcinoma cells by RT-PCR (unpublished data). Our study revealed that the CD24 mRNA activity was significantly lower in Hct116 less than HT29 cell (Figure 4C). β-actin was the internal control.
Shapira, S.; Shapira, A.; Starr, A.; Kazanov, D.; Kraus, S.; Benhar, I.; Arber, N. An immunoconjugate of anti-cd24 and pseudomonas exotoxin selectively kills human colorectal tumors in mice. Gastroenterology 2011, 140, 935-946. 10.1053/j.gastro.2010.12.004.

Reviewer 2 Report
Comment on the manuscript cells-564403 by Chen, et al.
Death domain-associated protein (DAXX) is known to function as a potent transcription repressor in the nucleus, and as a regulator of apoptosis in the cytoplasm. The authors investigated that the expression level of DAXX was decreased in CRC samples by western blotting and that this molecule modulates the proliferation of CRC cells. Although the paper is interesting, it also includes several problems as described below.
1) In Figure 1, the authors found that DAXX expression had inverse correlation with ß-catenin expression, and concluded that DAXX works as suppressor by repressing the WNT pathway. However, as well-known, ß-catenin exists on the cell membrane as well as the nucleus, and therefore this data could not demonstrate that DAXX negatively affects the cell proliferation. Instead, it should be determined the correlation of DAXX expression and localization of ß-catenin.
2) In Figure 3 and 4, it can be found that DAXX stimulates CD24 expression, and positively affect proliferation of CRC cells, however, not to be concluded that inactivation of DAXX promotes cell growth of CRC cells via CD24 downregulation. Then it needs other 2 experiments; (1) CD24 would conversely affect the DAXX expression or not, and (2) CD24 knock-down could affect the CRC proliferation or not.
3) In Figure 4, why the authors did not investigate the CD24 expression by shDAXX in HT29 cells?
4) It is unclear about the biological meanings of DAXX expression and CEA levels. It is well known that CEA is one of the biomarkers for progression of CRC. For example, serum CEA level exceeds 10ng/mL in most CRC patients with liver or lung metastasis but not in those without them. Therefore, DAXX expression is associated with CRC proliferation as indicated by the authors, that reflected inverse correlation of DAXX expression and serum CEA levels.
Author Response
Comments from Reviewer 2
Comment on the manuscript cells-564403 by Chen, et al.
Death domain-associated protein (DAXX) is known to function as a potent transcription repressor in the nucleus, and as a regulator of apoptosis in the cytoplasm. The authors investigated that the expression level of DAXX was decreased in CRC samples by western blotting and that this molecule modulates the proliferation of CRC cells. Although the paper is interesting, it also includes several problems as described below.
Comment 1: In Figure 1, the authors found that DAXX expression had inverse correlation with ß-catenin expression, and concluded that DAXX works as suppressor by repressing the WNT pathway. However, as well-known, ß-catenin exists on the cell membrane as well as the nucleus, and therefore this data could not demonstrate that DAXX negatively affects the cell proliferation. Instead, it should be determined the correlation of DAXX expression and localization of ß-catenin.
Response 1: Thank you for this comment. It would have been interesting to explore this aspect. Although, in the CRC samples demonstrate significantly negative correlation between the DAXX expression and the β-catenin expression (rho= −0.276, p<0.005; Figure 1C). Based on the results presented in vitro, cell proliferation increased by 1.77-fold at 60 h in DAXX-knockdown Hct116 cells compared with negative control. These data clearly demonstrate that DAXX knockdown significantly increased proliferation of Hct116 cells compared with that of 293T cells (Figure 2). Furthermore, we provide DAXX did not change β-catenin localization in DAXX-knockdown Hct116 cell by immunofluorescence analysis. As shown in figure below, we confirmed the β-catenin protein were present (green) on the cell membrane in DAXX-knockdown or luciferase-knockdown of Hct116 cells.
Comment 2: In Figure 3 and 4, it can be found that DAXX stimulates CD24 expression, and positively affect proliferation of CRC cells, however, not to be concluded that inactivation of DAXX promotes cell growth of CRC cells via CD24 downregulation. Then it needs other 2 experiments; (1) CD24 would conversely affect the DAXX expression or not, and (2) CD24 knock-down could affect the CRC proliferation or not.
Response 2: Thank you for pointing this out. Because of we previously analyzed gene microarray expression to identify DAXX-regulated genes and found that DAXX affects CD24 expression in Hct116 cells (unpublished data). Thus, we investigated DAXX as an important repressor in controlling CRC cell proliferation and migration (Figures 2 and 3). Taken together, DAXX downregulation attenuates CD24 mRNA and protein expression in Hct116 cells (Figure 4B and C). Similarly, in our study, DAXX expression was positive correlated with CD24 expression in malignant patients with CRC. In addition, CD24 may be regulated through Wnt signaling and may enhance CRC cells’ colony-forming ability and promote cell motility. CD24, proposed as a β-catenin target in the CRC cells, is expressed in breast cancer cells [35,36]. Taken together, DAXX may act as a potential repressor by controlling downstream CD24 expression in patients with CRC.
References:
Shulewitz, M.; Soloviev, I.; Wu, T.; Koeppen, H.; Polakis, P.; Sakanaka, C. Repressor roles for tcf-4 and sfrp1 in wnt signaling in breast cancer. Oncogene 2006, 25, 4361-4369. 10.1038/sj.onc.1209470. Ahmed, M.A.; Jackson, D.; Seth, R.; Robins, A.; Lobo, D.N.; Tomlinson, I.P.; Ilyas, M. Cd24 is upregulated in inflammatory bowel disease and stimulates cell motility and colony formation. Inflammatory bowel diseases 2010, 16, 795-803. 10.1002/ibd.21134.
Comment 3: In Figure 4, why the authors did not investigate the CD24 expression by shDAXX in HT29 cells?
Response 3: Thank you for pointing this out. Previous study indicated that the human colorectal adenocarcinoma cell line HT-29 and Hct116 cells can be CD24-positive and CD24-negative control for CRC, respectively [40]. In additional, we previously analyzed gene microarray expression to identify DAXX-regulated genes and found that DAXX affects CD24 expression in Hct116 cells (unpublished data). So that, we just take HT-29 cell as a CD24-positive control to compare CD24 expression of the Hct116 cells.
References:
Shapira, S.; Shapira, A.; Starr, A.; Kazanov, D.; Kraus, S.; Benhar, I.; Arber, N. An immunoconjugate of anti-cd24 and pseudomonas exotoxin selectively kills human colorectal tumors in mice. Gastroenterology 2011, 140, 935-946. 10.1053/j.gastro.2010.12.004.
Comment 4: It is unclear about the biological meanings of DAXX expression and CEA levels. It is well known that CEA is one of the biomarkers for progression of CRC. For metastasis but not in those without them. Therefore, DAXX expression is associated with CRC proliferation as indicated by the authors, that reflected inverse correlation of DAXX expression and serum CEA levels.
Response 4: Thank you for pointing this out. Clinically, serum CEA levels are mainly used as a tumor marker to monitor colorectal carcinoma treatment and identify recurrences after surgical resection [56]. However, a lack of sensitivity renders its use limited in clinical diagnosis (PMID: 26035703). Thus, it is imperative to identify novel biomarkers and develop novel treatment strategies for CRC. An investigation to determine the DAXX expression was significantly lower in the patients with the positive serum (CEA) screening results compared to the patients with negative CEA screening levels (p < 0.001) (Table 1). Here, we further revealed that the DAXX expression at low levels had significantly correlated with CD24 expression in CEA-positive subgroup (rho = 0.461, p < 0.005) (Figure 1E). Therefore, we clarified if DAXX plays a role in CRC carcinogenesis and can provide a reference for further therapeutic targets in CRC.
References:
Cai, Z.; Xiao, J.; He, X.; Ke, J.; Zou, Y.; Chen, Y.; Wu, X.; Li, X.; Wang, L.; Wang, J., et al. Accessing new prognostic significance of preoperative carcinoembryonic antigen in colorectal cancer receiving tumor resection: More than positive and negative. Cancer biomarkers : section A of Disease markers 2017, 19, 161-168. 10.3233/cbm-160287.
PMID: 26035703
Thomas, D.S.; Fourkala, E.O.; Apostolidou, S.; Gunu, R.; Ryan, A.; Jacobs, I.; Menon, U.; Alderton, W.; Gentry-Maharaj, A.; Timms, J.F. Evaluation of serum cea, cyfra21-1 and ca125 for the early detection of colorectal cancer using longitudinal preclinical samples. British journal of cancer 2015, 113, 268-274. 10.1038/bjc.2015.202
